# Microbiota composition effect on immunotherapy outcomes in colorectal cancer patients: A systematic review

**Suad Mohamed Ajab**[1], **Sumaya Hasan Zoughbor**[2], **Lena Abdulbaset Labania**[ID][2], **Linda Mari Östlundh**[3], **Hiba Salaheldin Orsud**[2], **Marie Antonette Olanda**[2], **Obaid Alkaabi**[2], **Shamma Hamad Alkuwaiti**[2], **Shaikha Mohammed Alnuaimi**[2], **Zakeya Al Rasbi**[ID][2]*

**1** Institute of Public Health, College of Medicine and Health Sciences, United Arab Emirates University, Al Ain, Abu Dhabi, United Arab Emirates, **2** Microbiology and Immunology, College of Medicine and Health Sciences, United Arab Emirates University, Al Ain, Abu Dhabi, United Arab Emirates, **3** University Library, Örebro University, Örebro, Sweden

* rasbi@uaeu.ac.ae

**Data Availability Statement:** ll relevant data are within the manuscript and its Supporting Information files.

## Abstract

### Background

Immune checkpoint inhibitors (ICIs) have emerged as an effective treatment for colorectal cancer (CRC). Studies indicate that the composition of gut microbiota could potentially serve as a biomarker for predicting the clinical effectiveness of immune checkpoint inhibitors.

### Methods

Following PRISMA guidelines, the review was conducted after registering the protocol with PROSPERO. A comprehensive literature search was carried out across five databases: PubMed, Scopus, Web of Science, Embase, and Cochrane Library. Assessment tools from the National Institutes of Health (NIH) were used to gauge the quality of the studies

### Results

A total of 5,132 papers were identified, and three studies and one conference abstract published between 2017–2022 met the inclusion criteria and were summarized in a descriptive synthesis table. These four studies were in accord with the following findings, four main phyla, *Firmicutes*, *Bacteroidata*, *Actinobacteria*, and *Verrucomicrobiota* were associated with CRC patients' clinical response toward ICIs treatment. *Ruminococcaceae* was predominantly related to CRC patients responding to therapy, while the *Micrococcaceae* family was more common among the non-responders. Bacterial taxa such as *Faecalibacterium* and *Prevotellaceae* were associated with better responses to ICIs and could be predictive biomarkers. The signature of fecal microbiota with *Akkermansia muciniphila* and *Eubacterium rectale* enrichment, and *Rothia mucilaginosa* depletion could independently predict better response to ICIs in patients with CRC.

**Funding:** United Arab Emirates University, College of Graduate Studies, Ph.D. grant number 12M081 supported this work resources (https://www.uaeu.ac.ae/en/cgs/scholarship.shtml). The sponsor play no role in the study design, data collection or analysis, decision to publish or preparation of the manuscript.

**Competing interests:** The authors have declared that no competing interests exist.

## Conclusion

The findings have brought attention to the notable differences in terms of richness and composition of microbiota between patients who responded positively to the treatment and those who did not. Bacterial species and families, such as *Faecalibacterium*, *Bifidobacterium*, *Lachnospiraceae*, *Akkermansia sp.*, *Ruminococcaceae*, *and Prevotellaceae*, have consistently surfaced as potential indicators of immunotherapeutic responses. Furthermore, this review also emphasizes the need for additional comprehensive, multi-center studies with larger sample sizes to validate reported microbiota and expand our understanding of the role of gut microbiota in CRC ICIs therapy.

PROSPERO ID: CRD42021277691

## Introduction

### Background

Cancer is a major contributor to global mortality, with colorectal cancer ranking as the second leading cause of cancer-related deaths in 2020, according to the Global Cancer Epidemiological Statistics (GCES) released by the World Health Organization (WHO) [1, 2].

Growing evidence indicates that the composition of humans' gut microbiota significantly determines our overall health and response to disease [3, 4]. The composition of the gut microbiota, consisting of various microorganisms such as bacteria, archaea, viruses, fungi, and protozoa, has been linked to the emergence and progression of gastrointestinal cancers like colorectal cancer [5, 6]. Recent studies have indicated a significant relationship between the gut microbiota and the efficacy of cancer therapy [1]. The field of cancer immunotherapy is advancing rapidly, with increasing attention to the reciprocal influence between the gut microbiome and cancer treatment outcomes; thus, investigating the role of the gut microbiome in anti-cancer immune surveillance and immunotherapy holds great promise in optimizing treatment responses in cancer patients [7, 8]. The human gut microbiota composition consists primarily of two major phyla: *Bacteroidetes* and *Firmicutes* [3]. Various factors such as diet, culture, residence, and medications influence the makeup of the gut microbiota [4, 9]. However, longitudinal studies have shown that individuals who respond favorably to cancer treatment regimens are loaded in specific bacterial taxa than those who do not respond well [10, 11].

The immune system has an essential role in controlling cancer growth and progression. Tumor cells can evade immune system recognition by multiple mechanisms. Immunotherapy augments the human immune defense against cancer. The immune checkpoints pathway encompasses various proteins that exert activating and inhibitory effects to maintain immunological self-tolerance. These checkpoints are instrumental in regulating cytotoxic T-lymphocyte activation, preserving self-tolerance, preventing autoimmunity, and modulating the intensity and duration of immune reactions to limit inflammation-induced tissue damage [12].

The most well-described checkpoints are the cytotoxic T lymphocyte-associated molecule-4 (CTLA-4), programmed cell death receptor-1 (PD-1), and programmed cell death ligand-1 (PD-L1). Monoclonal antibodies block these pathways to enhance the host immunologic activity against tumors and become the standard of care for many malignancies [12]. Currently, the immune checkpoint inhibitors approved by the FDA for CRC are anti-(CTLA-4) (ipilimumab), PD-1 programmed cell death protein 1 (pembrolizumab and nivolumab), and PDL-1 (atezolizumab, avelumab, and durvalumab). [11–13]

Several studies have investigated gut microbiota's impact on cancer patients' response to immune checkpoint inhibitors [9–11, 13]. However, limited data is available to ascertain whether gut microbiota can serve as a reliable biomarker for assessing the effectiveness of ICIs in colorectal cancer therapy [14].

The present systematic review seeks to build upon the findings from recent systematic reviews in 2021, one of which evaluated the efficacy of ICIs in treating advanced solid cancers, including CRC, and found that combined therapy significantly improved disease control rates and survival time at the cost of increased severe adverse reactions compared to monotherapy [15]. The other systematic review explored the potential of tumor mutational burden as a predictive biomarker for CRC patients receiving ICIs therapy [16]. With these foundations, our systematic review aims to extend these understandings focusing on the association between gut microbiota and the clinical efficacy of ICIs in CRC patients.

## Methods

We conducted the systematic review following the 2020 guideline of the Preferred Reporting Items for Systematic Review and Meta-Analyses (PRISMA) statement, **S1 and S2** **Tables** [17]. The review protocol was developed and registered with the International Prospective Register of Systematic Reviews Platform (PROSPERO) under registration ID CRD42021277691 [18].

### Study selection and inclusion criteria

We conducted a thorough literature search in five biomedical databases, including PubMed, Scopus, Web of Science, Embase, and Cochrane Library, until July 18, 2023, **S2 Table**. The search terms used were: (Gastrointestinal Microbiome OR Gastrointestinal Microbiota OR Gastrointestinal Microflora OR Gastrointestinal Flora OR Gut Microbiome OR Gut Microbiota OR Gut Microflora OR Gut Flora OR Intestinal Microbiome OR Intestinal Microbiota OR Intestinal Microflora OR Intestinal Flora) AND (nivolumab OR pembrolizumab OR atezolizumab OR avelumab OR durvalumab OR ipilimumab OR tremelimumab OR immunotherapy OR PD-1 OR PD-L1 OR CTLA-4 OR immune checkpoint OR checkpoint blockade OR immune checkpoint inhibitors OR ICI OR ICIs OR immune checkpoint blockers OR ICB OR ICBs) AND (colon cancer OR colorectal cancer OR colorectal carcinoma OR colon carcinoma OR colonic neoplasm OR colorectal neoplasm) (**S1 Table**). The study protocol provides a detailed description of the search strategy and has been made publicly accessible [18].

Reviewers (SA, SZ, MO, LL, and ZA) conducted the study selection and evaluation process via Covidence (Melbourne, Australia) Disagreements were resolved through discussion and consensus between (SA, SZ, and ZA). Duplicate publications were removed before applying five criteria in the PICOS format to select the articles. These criteria included: (1) Participants included patients of any age and sex who were diagnosed with colorectal cancer; (2) The intervention involved the use of ICIs for cancer immunotherapy, such as anti-PD-1, anti-PD-L1, anti-CTLA-4, or a combination of ICIs at various doses and timings, (3) The comparison focused on the gut microbiota composition reported from molecular analysis in CRC patients, (4) The outcomes were evaluated based on the classification of patients into "responders" and "non-responders" to ICIs, following the RECIST 1.1 criteria. and (5)

In our systematic review, we concentrated on published observational human studies that had been subjected to the peer-review process, ensuring the inclusion of only the most robust and directly applicable data. We excluded experimental animal studies, recognizing the challenges in translating such findings to human physiology. Additionally, reviews, comments, case reports, editorials, and conference abstracts were omitted due to their lack of original empirical research, which is essential for the integrity of a rigorous systematic review. This

approach was designed to furnish the clinical oncology community with an analysis grounded in reliable and broadly applicable evidence.

Despite our initial assertion to exclude conference abstracts, we made an exception for one abstract from a large-scale clinical trial (Checkmate 142) [19]. Considering the nascent nature of the systematic review topic and the limited number of clinical studies available, including this abstract was deemed necessary to augment data synthesis. Additionally, a chosen study incorporated mouse experiments [20]. Its clinical importance and the limited number of full-fledged studies on CRC patients in this niche area warranted its inclusion. Consequently, after evaluating the full texts, we selected studies that fulfilled the pre-determined eligibility criteria.

## Data extraction and quality assessment

The selected studies were reviewed (SA, SZ, HO, ZR, OA, SHKU, SHNU), and data was extracted independently by (SA, SZ, HO, and ZA). The collected data consisted of the author's information, publication year, number of patients, ICI therapy type, patient recruitment location, gut microbiota sequencing and analysis methods used, outcomes related to diversity and taxonomy of the gut microbiota, as well as survival outcomes as in responders versus non-responders to cancer ICIs therapy. The included studies consisted of two prospective cohort studies, one case-control, and a snapshot (cross-sectional) analysis from a randomized controlled trial (**Table 1**).

To assess the rigor of the studies, we utilized standardized assessment tools from the National Institutes of Health (NIH) that align with each specific study type (**Table 2**). These tools provided scores ranging from 0 to 14. Studies were categorized as having intermediate quality if they scored between 5 and 11, while those scoring between 12 and 14 were deemed to have good quality [21].

## Ethics and dissemination

Ethics approval for the current systematic review methodology is not required due to the nature of the study design.

## Results

### Literature search results

After utilizing the systematic review software Covidence [22] to eliminate 1,887 duplicated articles, we identified 5,132 papers from PubMed, Scopus, Web of Science, Embase, and Cochrane Library databases. After thoroughly examining the full texts, we excluded 19 studies deemed irrelevant to the topic or did not meet the criteria for proper comparison, indication, intervention, study design, or patient population. Ultimately, four studies were included in our analysis (**Fig 1**).

### Characteristics of studies

Four studies have met the criteria for examining the gut microbiota in patients with colorectal cancer who underwent ICIs cancer therapy. These studies were published between 2017 and 2022 and involved sample sizes ranging from 60 to 89 participants. Cheng et al. 2022 [23], Peng et al. 2020 [24], and Pi et al. 2020 [20] conducted their studies in China. On the other hand, the study by Koptez et al. 2017 [19] is a sub-study of CheckMate 142, an international clinical trial conducted in eight countries: Australia, Belgium, Canada, France, Ireland, Italy, Spain, United States, registered under ClinicalTrials.gov identifier NCT02060188 (**Table 1**).

**Table 1. Clinical studies evaluating the association between gut microbiome and clinical efficacy of the ICIs.**

| Publication characteristics | | | Study design | | | Patients' characteristics | Intervention characteristics | | Clinical outcome | Statistical analysis | NIH Quality Assessment Tool score* |
|---|---|---|---|---|---|---|---|---|---|---|---|
| First author (Publication year)/DOI | Country | Study type | Dates of enrolment | Inclusion and exclusion criteria | Total number of enrolled patients | | Type of cancer therapy/ Duration | Measurement of exposure | | | |
| Kopetz* (2017)/ DOI: 10.1186/ s40425-017-0288-4 | International‡ multi-center clinical trial | Cross-sectional (sub-study from a clinical trial) | NCT02060188 | **Inclusion:** • Age 18 years and older • CRC patients with dMMR/ MSI-H enrolled in CheckMate 142 | 72 metastatic CRC | | Nivolumab ± Ipilimumab/ every 2 or 3 wks | 16S rRNA-sequencing | • Responders = 25 (PR) • Non-responders = 14 • (PD) | Due to a limited number of patients, nivolumab and combined nivolumab and ipilimumab treatments were grouped together, potentially affecting precise microbiota-response relationships for each ICIs regimen. | 6 |
| Pi (2020)/ DOI: 10.3233/ CBM-201606 | China | Case-control | Jan 2018- Jan 2019 | **Inclusion:** • 30 CRC patients treated with PD-1 monoclonal antibody therapy (pembrolizumab) • 30 CRC patients treated with routine non-immune therapy (chemotherapy) | 60 CRC | | • Cases: Pembrolizumab/ every 2 to 3 weeks • Controls: Chemotherapy/ NM | • 16S rRNA-V4-V3 amplicon sequencing • Metagenomic whole-genome shotgun sequencing | • Responders PR 70% • SD 55% • Non responders (NM) • PD | • Software Name: SPSS 17.0 and GraphPad Prism 5.0 • Outline: Normality tests were initially performed on the data. If data met normality, ANOVA and pairwise comparisons were done. Non-normal data was analyzed with Kruskal-Wallis H test, and depending on variance homogeneity, either a post-hoc LSD test or Dunnett's test was applied. | 7 |

*(Continued)*

**Table 1.** (Continued)

| Publication characteristics | | Study design | | | Patients' characteristics | Intervention characteristics | | Clinical outcome | Statistical analysis | NIH Quality Assessment Tool score* |
|---|---|---|---|---|---|---|---|---|---|---|
| First author (Publication year)/DOI | Country | Study type | Dates of enrolment | Inclusion and exclusion criteria | Total number of enrolled patients | Type of cancer therapy/ Duration | Measurement of exposure | | | |
| Peng (2020)/ DOI: 10.1158/ 2326-6066. CIR-19-1014 | China | Single Center Prospective cohort | Feb 2017-Jan 2018 | **Inclusion:** • Patients with gastrointestinal (GI) cancer • Receiving anti–PD-1/PD-L1 treatment OR anti–PD-1/ PD-L1 treatment combined with anti CTLA-4 **Exclusion:** • Patients who received combined chemotherapy and anti–PD-1/ PD-L1 immunotherapy | • 89 GI cancer patients enrolled • 74 patients provided samples (19 CRC) | Anti–PD-1/PD-L or combined with Anti-CTLA-4/ every 2 or 3 wks | • 16S rRNA-V3-V4 amplicon sequencing N = 74 • Metagenomics shotgun sequencing N = 55, • Metagenomics analysis N = 40 | • Responders = 45 Of which CRC = 12 • (PR/SD) • Non-responders = 29 • Of which CRC = 7 • (PD) | • Software Name: R software • Outline:1. Kaplan-Meier method: Employed for estimating progression-free survival.2. Log-rank tests: Utilized to compare progression-free survival data.3. Kolmogorov-Smirnov tests: Used for inspecting the distribution ratio between two bacterial types among patients.4. Pearson coefficient: Applied for analyzing correlations between quantities of common microbial genera discovered via two sequencing methods (16S rRNA and metagenomics). | 12 |

**Table 1.** (Continued)

| Publication characteristics | | Study design | | | Patients' characteristics | Intervention characteristics | | Clinical outcome | Statistical analysis | NIH Quality Assessment Tool score* |
|---|---|---|---|---|---|---|---|---|---|---|
| First author (Publication year)/DOI | Country | Study type | Dates of enrolment | Inclusion and exclusion criteria | Total number of enrolled patients | Type of cancer therapy/ Duration | Measurement of exposure | | | |
| Cheng (2022)/ DOI: 10.3390/ jcm11185479 | China | Single center Prospective cohort | Sep 2019-April 2020. | **Inclusion:** • Aged ≥ 18 years • Histologically confirmed cancer, irrespective of tissue origin • No previous immunotherapy taken • No synchronous or metachronous cancer • No antibiotic treatment taken in the preceding 2 months **Exclusion:** • Patients with active progressing brain metastases • History of serious autoimmune disease | 72 patients advanced cancer III–IV stage (5 CRC) | • Nivolumab/ every 2 weeks • Pembrolizumab, Sintilimab, Camrelizumab and Toripalimab/ every 3 weeks | 16S rRNA-V4 amplicon pyrosequencing | • Responders = 33 Of which CRC = 1 • (PR/SD) • Non-responders = 39 • Of which CRC = 4 • (SD/PD) | • Software Name: SPSS software (version 23.0). • Outline:1. Categorical Baseline Variables: Fisher's exact test or Chi-squared test used for comparisons.2. Continuous Baseline Variables: t-test was employed for comparisons.3. Associations between microbiota profiles and immunological parameters: Spearman's correlation coefficient and the two-sided Wilcoxon test were utilized. | 11 |

*Poster 392

‡ Australia, Belgium, Canada, France, Ireland, Italy, Spain, and the United States

**CRC:** Colorectal cancer

**NM:** Not mentioned

**PR:** Partial response, SD: Stable disease PD: Progressive disease

**dMMR/MSI-H:** Deficient DNA mismatch repair/microsatellite instability-high

*Total score (/14): Fair (5–10), Good (11–14)

**Table 2. Risk of bias for included studies using NIH quality assessment tool.**

| a | National Institutes of Health Quality Assessment Tool | Peng, 2020 | Kopetz, 2017 | Cheng 2022 |
|---|---|---|---|---|
| 1 | Was the research question or objective in this paper clearly stated? | Y | Y | Y |
| 2 | Was the study population clearly specified and defined? | Y | Y | Y |
| 3 | Was the participation rate of eligible persons at least 50%? | Y | NA | Y |
| 4 | Were all the subjects selected or recruited from the same or similar populations (including the same time period)? Were inclusion and exclusion criteria for being in the study prespecified and applied uniformly to all participants? | Y | NA | Y |
| 5 | Was a sample size justification, power description, or variance and effect estimates provided? | N | NA | N |
| 6 | For the analyses in this paper, were the exposure(s) of interest measured prior to the outcome(s) being measured? | Y | Y | Y |
| 7 | Was the timeframe sufficient so that one could reasonably expect to see an association between exposure and outcome if it existed? | Y | NA | Y |
| 8 | For exposures that can vary in amount or level, did the study examine different levels of the exposure as related to the outcome (e.g., categories of exposure, or exposure measured as continuous variable)? | Y | Y | Y |

(*Continued*)

**Table 2.** (Continued)

| a | National Institutes of Health Quality Assessment Tool | Peng, 2020 | Kopetz, 2017 | Cheng 2022 |
|---|---|---|---|---|
| 9 | Were the exposure measures (independent variables) clearly defined, valid, reliable, and implemented consistently across all study participants? | Y | Y | Y |
| 10 | Was the exposure(s) assessed more than once over time? | Y | N* | Y |
| 11 | Were the outcome measures (dependent variables) clearly defined, valid, reliable, and implemented consistently across all study participants? | Y | Y | Y |
| 12 | Were the outcome assessors blinded to the exposure status of participants? | NR | NR | NR |
| 13 | Was loss to follow-up after baseline 20% or less? | Y | NA | Y |
| 14 | Were key potential confounding variables measured and adjusted statistically for their impact on the relationship between exposure(s) and outcome(s)? | Y | NA | N |
| **Total score (/14)** | | **12** | **6** | **11** |
| **Fair** (5–10 out of 14 questions) | | | | |
| **Good** (11–14 out of 14 questions) | | | | |

| b | National Institutes of Health Quality Assessment Tool | Pi, 2020 |
|---|---|---|
| 1 | Was the research question or objective in this paper clearly stated and appropriate? | Y |
| 2 | Was the study population clearly specified and defined? | Y |
| 3 | Did the authors include a sample size justification? | N |
| 4 | Were controls selected or recruited from the same or similar population that gave rise to the cases (including the same timeframe)? | Y |
| 5 | Were the definitions, inclusion and exclusion criteria, algorithms or processes used to identify or select cases and controls valid, reliable, and implemented consistently across all study participants? | Y |
| 6 | Were the cases clearly defined and differentiated from controls? | Y |
| 7 | If less than 100 percent of eligible cases and/or controls were selected for the study, were the cases and/or controls randomly selected from those eligible? | NR |

(*Continued*)

**Table 2.** (Continued)

| a | National Institutes of Health Quality Assessment Tool | Peng, 2020 | Kopetz, 2017 | Cheng 2022 | |
|---|---|---|---|---|---|
| 8 | Was there use of concurrent controls? | | | | CD |
| 9 | Were the investigators able to confirm that the exposure/risk occurred prior to the development of the condition or event that defined a participant as a case? | | | | Y |
| 10 | Were the measures of exposure/risk clearly defined, valid, reliable, and implemented consistently (including the same time period) across all study participants? | | | | Y |
| 11 | Were the assessors of exposure/risk blinded to the case or control status of participants? | | | | NR |
| 12 | Were key potential confounding variables measured and adjusted statistically in the analyses? If matching was used, did the investigators account for matching during study analysis? | | | | N |
| **Total score (/14)** | | | | | 7 |
| **Fair** (5–10 out of 14 questions) | | | | | |
| **Good** (11–14 out of 14 questions) | | | | | |

a. Risk of bias for included studies using NIH Quality Assessment Tool for Observational Cohort and Cross-sectional Studies

b. Risk of bias for included studies using NIH Assessment of Case-Control Studies

**Y:** yes, **N:** no

**CD**: cannot determine **NA:** not applicable **NR:** not reported

Cheng et al. and Peng et al. were prospective cohort studies that included 72 and 89 patients, 5 and 19 of whom had CRC, respectively [19, 20]. The third study, Pi et al., involved a case-control design that examined 60 patients with colorectal cancer who underwent various treatment regimens, including immune checkpoint inhibitors [20]. Kopetz et al. enrolled a total of 72 patients with colorectal cancer [19].

Studies conducted by Cheng et al., Peng et al., and Kopetz et al. examined the relationship between gut microbiome composition and the response to anti-PD-1/PD-L1 immunotherapy in colorectal cancer patients following the RECIST 1.1. In Pi et al.'s analysis, bacterial fluctuations were noted; however, the association with anti-PD-1/PD-L1 immunotherapy response in CRC patients was not distinctly articulated. Although the study suggested a correlation, the inadequate labeling and explanation in the figures constrained the interpretation of the data. Consequently, the results were inferred by contrasting shifts in bacterial taxa within both groups, drawing on previous studies that utilized similar bioinformatics and visualization tools for comparison.

## Predictive taxonomic profiles of CRC patients in response to ICIs

The taxonomic distribution or differential abundance of microbial communities differed between CRC patients responding to immunotherapy and non-responders.

Kopetz and colleagues [19] investigated colorectal cancer patients who had undergone ICIs, nivolumab, and ipilimumab and possessed deficiency in DNA mismatch repair (dMMR), causing high microsatellite instability (MSI-H). The study observed shifts in the gut microbiome using 16S rRNA sequencing on baseline stool samples. This investigation was a component of a broader clinical trial series, CheckMate 142. Remarkably, *Rothia* and *Rothia mucilaginosa*, from the *Micrococcaceae* family within the *Actinobacteria* phylum, exhibited a 2.7-fold and 3-fold lower abundance, respectively, in patients partially responding to CRC ICIs compared to patients with progressive disease (**Figs 2 and 3**) [19].

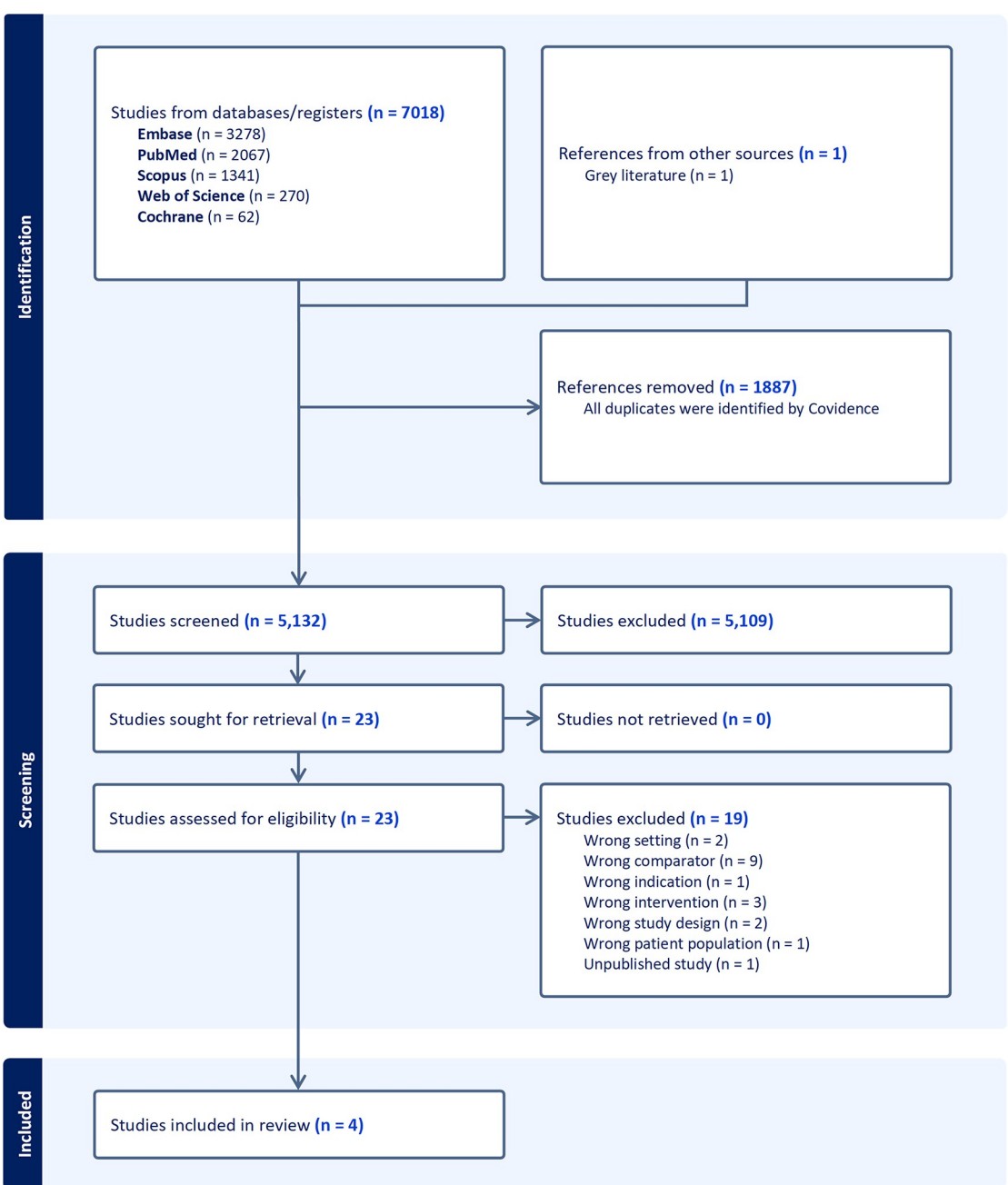

**Fig 1. PRISMA flowchart and decision-making process for the inclusion criteria of studies.**

Pi and colleagues [20] performed 16S rRNA sequencing on CRC patients' fecal specimens at the time of diagnosis to identify microbes in the patients' gut. The researchers investigated the correlation between the number of genes (gene count) or the quantity of these identified metagenomic species (MGS) abundance and the patients' 3-month progression-free survival. The term' 3-month PFS' refers to the time after treatment initiation, during and after which a patient lives without the disease worsening. In this instance, the researchers found that they could not predict the 3-month PFS based on the gene count or the abundance of MGS.

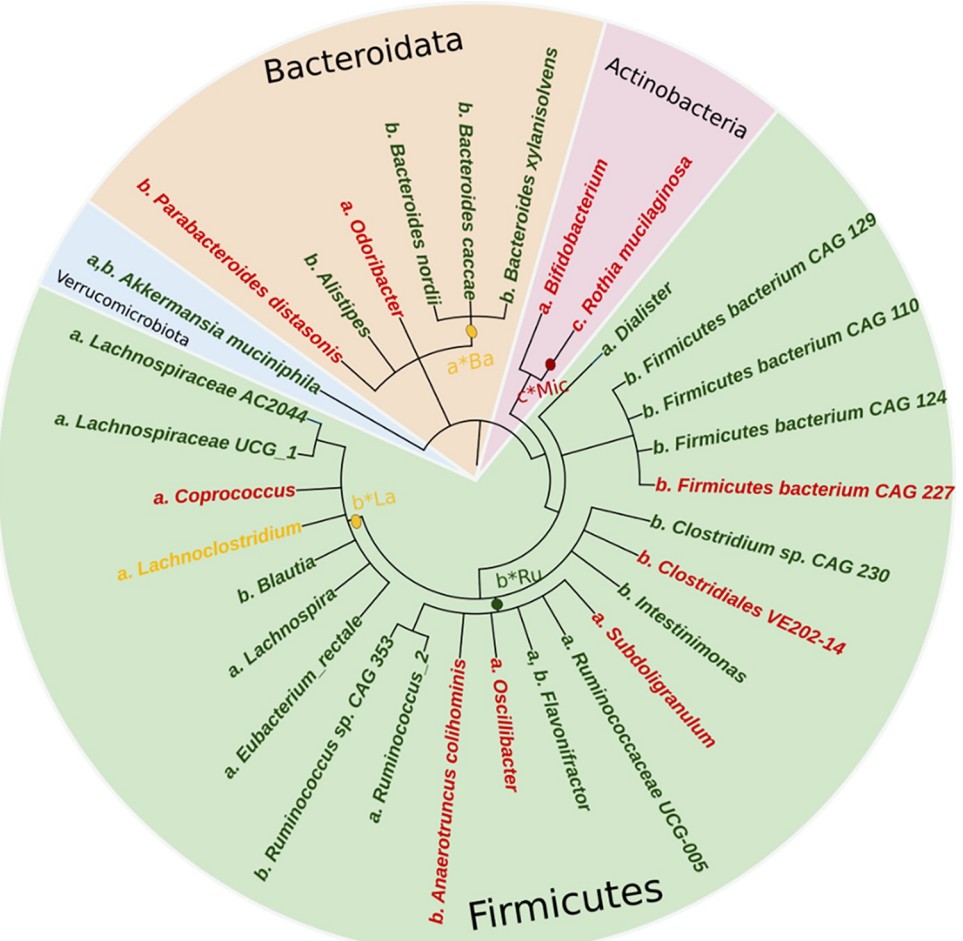

**Fig 2. Phylogenetic representation of gut microbiome association with ICIs treatment response in CRC patients.**
The tree was constructed via phyloT software and visualized using iTOL enabling a comprehensive classification of
bacterial taxa identified in the systematic review, presented in an inward manner from phylum to species.
Differentially enriched bacterial taxa associated to better response were labelled in **green** while those associated with
poorer response are in **red**. Taxa with mixed findings are denoted in **yellow** (i.e. in responder and non-responder
groups). Lowercase alphabets (a, b, and c) adjacent to each bacterial taxon indicate the source study, asterisk (*)
indicates identification at the family taxonomic level (S6 Table).

According to Pi et al.'s findings, CRC ICIs responders with six months of progression-free
survival demonstrated a high abundance of the *Akkermansia muciniphila*, from the *Verrucomi-
crobiota* phylum, in addition to a unique mix of co-abundant bacterial gene (CAG) groups.
These include, but are not limited to, *Alistipes*, *Intestinimonas*, *Bacteroides nordi*, *Bacteroides
xylanisolvens*, *Blautia*, *Lachnospiraceae*, *Firmicutes bacterium* CAG:129, and *Ruminococcus sp.*
CAG:353 (**Fig 2 and S6 Table**).

Conversely, CRC ICIs non-responders were found to harbor a different set of bacterial
CAGs that included *Clostridiales*, *Parabacteroides distasonis*, *Firmicutes bacterium* CAG:227,
and *Anaerotruncus colhominis* (**Figs 2 and 3**) [20].

Peng et al. [24] implemented 16S rRNA gene sequencing to investigate the link between gut
microbiota and the clinical effectiveness of ICIs in advanced-stage gastrointestinal cancer
patients. Their research focused on identifying differential abundant Operational Taxonomic
Units (OTU) among responders and non-responders to ICIs. In CRC patients, the *Prevotella-
ceae* in the responder group were not significantly abundant; however, there was more

**ICIs Responders**

- ↓ *Micrococcaceae*
- • *Rothia Mucilaginosa*
- ↑ *Bacteroides*
- • *Bacteroides nordii*
- • *Bacteroides caccae*
- • *Bacteroides xylanisolvens*
- ↑ *Parabacteroides*
- ↑ *Lachnospiraceae*
- • *Lachnospira*
- • *Lachnoclostridium*
- ↑ *Clostridiales*
- • *Flavonifractor*
- ↑ *Firmicutes bacterium*
- ↑ *Eubacterium rectale*
- ↑ *Akkermansia muciniphila*
- ↑ *Intestinimonas*
- ↑ *Blautia*
- ↑ *Dialister*
- ↑ *Alistipes*
- ↑ *Ruminococcus*

**ICIs Non-responders**

- ↓ *Bacteroides*
- ↓ *Parabacteroides*
- • *Parabacteroides distasonis*
- ↓ *Lachnospiraceae*
- • *Lachnoclostridium*
- ↓ *Firmicutes bacterium*
- ↑ *Micrococcaceae*
- • *Rothia Mucilaginosa*
- ↑ *Clostridiales*
- ↑ *Coprococcus*
- ↑ *Subdoligranulum*
- ↑ *Odoribacter*
- ↑ *Oscillibacter*
- ↑ *Faecalibacterium*
- ↑ *Bifidobacterium*
- ↑ *Anaerotruncus*

**Fig 3. Comparison of bacterial taxonomy in ICIs-responsive vs non-responsive colorectal cancer patients.** This comparison is conducted regardless of the sequencing approach reported in Peng et al., Kopetz et al., and Pi et al. The arrows indicate whether each taxon is differentially enriched (up arrow) or less abundant (down arrow) in the patient groups. The image was created with mindthegraphBioRender.com. *ICIs-responsive vs non-responsive colorectal cancer patients © 2024 by Suad Ajab is licensed under CC BY 4.0*.

*Prevotellaceae* relative to the quantity of *Bacteroidaceae*. Interestingly, this pattern did not extend to gastric cancer patients, underlining unique gut microbiota interplays specific to cancer type. Also, an increased abundance of *Prevotellaceae* demonstrated greater chances of disease stabilization (PFS) within 12 weeks after ICIs therapy. Moreover, an increased abundance of bacterial taxa such as *Akkermansia muciniphila*, *Eubacterium rectale*, *Ruminococcaceae*, *Bacteroideceae*, and *Lachnospiraceae* were identified in ICIs responders. Conversely, CRC ICIs non-responders exhibited a higher abundance of *Bifidobacteria*, *Coprococcus*, and other OTUs within the *Ruminococcaceae* and *Bacteroides* family (**Figs 2 and 3**) [24].

A consistent correlation between *Prevotella* and *Bacteroides* was observed in both 16S rRNA sequencing data and metagenomics shotgun sequencing data, as reported by Peng et al. [24]

Lastly, using machine learning tools, Peng et al. aimed to predict patient responses to therapy based on the presence of bacteria. These predictive models, especially the ExtraTrees and elastic net methods, identified 15 of the top 20 predictors belonging to the *Firmicutes* phylum, indicating that they may serve as predictive markers in ICIs therapy effectiveness.

In a comprehensive study conducted by Cheng et al. [23] in 2022, they discovered notable differences in the gut microbial biodiversity of patients with advanced stages of various cancer types. However, they have provided a pooled summary of gut microbiota composition across all cancer types studied, and a data repository was not provided to access the specific microbiota variations in response to CRC ICIs therapy. The bacterial family *Prevotellaceae*, focusing on the genus *Prevotella*, was at a higher concentration in ICIs responders. Furthermore, Cheng and colleagues used the Linear discriminant analysis Effect Size (LefSe) statistical method to illustrate variances in the presence of microbial taxa within distinct groups. Prior ICIs therapy, pan-cancer responders tended to have an abundance of specific microbes such as *Archaea*, *Lentisphaerae*, *Victivallaceae*, *Victivallales*, *Lentisphaeria*, *Methanobacteriaceae*, *Methanobacteria*, *Euryarchaeota*, *Methanobrevibacter*, *Methanobacteriales*, and *Leuconostoc* (**S5 Table**). On the other hand, the pan-cancer non-responders predominantly exhibited a high presence of the *Clostridiaceae* family within the phylum *Firmicutes* [23]. Six weeks post treatment, the microbial profile among patients with various types of cancer changed; the responders had an increase in *Lachnospiraceae* and *Thermus*, while in non-responders, the levels of *Faecalibacterium* and *Bifidobacterium* significantly decreased (**S5 Table**) [23].

Cheng et al. carried out a network analysis on the gut microbiota, including 16 different bacterial phyla from 72 advanced cancer patients. This analysis aimed to determine how changes in the abundance of one bacterial group could affect the quantity of another. A positive correlation signals a simultaneous rise in the presence of two bacterial groups, while a negative correlation suggests an inverse relationship; an increase in one bacterium coincides with a reduction in another bacterium. For instance, *Bacteroidetes* were negatively correlated with *Firmicutes*, *Proteobacteria*, and *Actinobacteria*. Conversely, *Proteobacteria* and *Actinobacteria* displayed a positive correlation [23].

## Prediction of microbial functional pathways in response to ICIs

Peng et al. employed the HUMAnN2 tool to mine microbiota data for insights into associated metabolic and biological processes [24]. Advanced cancer patients responding to ICIs treatment exhibited enhanced bacterial activity associated with producing short-chain fatty acids, unsaturated fatty acids, vitamins, and starch. Indeed, the *Eubacterium rectale*, known for producing short-chain fatty acid and associated with fiber-rich diets, was vastly abundant in the CRC ICIs responsive group. Conversely, patients who responded poorly to ICIs showed elevated bacterial activity related to lipopolysaccharide production, sugar catabolism, and protein synthesis [24].

Using PICRUSt (Phylogenetic Investigation of Communities by Reconstruction of Unobserved States) function prediction, Cheng et al. found that metabolic pathways L3_Stilbenoid diaryl-heptanoid, gingerol biosynthesis, and the L3_Glycosaminoglycan degradation pathway was more prominent in responders before all cancer type ICIs treatment, **S4 Table** [23].

## Comparative analysis of methodologies utilized in microbiota profiling

The four studies [19, 20, 23, 24] used taxa-targeted methods focused on the 16S rRNA gene to identify bacterial presence in gut microbiota. The 2022 study by Cheng et al. utilized the 16S rRNA-V4 amplicon pyrosequencing method, which allows for genus-level classification but is unable to distinguish between species or strains of a genus in the 16S rRNA gene. Through this

method, an analysis was conducted on 9708 OTUs from 144 samples, identifying 10 genera without extending to the species level, **S3 Table** and **S1 Fig** [25].

The 2020 studies by Peng et al. and Pi et al. utilized 16S rRNA-V3-V4 amplicon sequencing. Though this method can identify and compare bacteria, it is limited in resolving bacterial taxa (see **S3 Table** and **S1 Fig** for more details). Nevertheless, applying metagenomics with 16S rRNA gene sequencing leads to high taxonomic resolution results, as observed in Peng et al.'s study, where the *Bacteroides* genus is present in both CRC responder and non-responder groups. However, a more refined perspective emerges when examining at the species level. For instance, Pi et al.'s metagenomic study identifies that specific *Bacteroides* species such as *Bacteroides nordii*, *Bacteroides caccae*, and *Bacteroides xylanisolvens* are associated with improved responses in CRC patients (**Fig 2** and **S6 Table**).

Peng et al.'s study utilized metagenomic shotgun sequencing on fecal samples from 55 out of the 74 advanced cancer patients. This method not only strengthened the initial 16S rRNA analysis findings but also introduced 10 additional differentially enriched species, see **S3 Table** and **S1 Fig**.

The applied methodologies in microbiota profiling and the choice of reference databases for taxonomic assignments can both introduce variation in the resolution of results obtained [26]. Cheng et al. and Pi et al. employed GreenGenes, while Peng used the SILVA database. GreenGenes, with almost 202,421 reference sequences, is widely used for its tree-based taxonomy placement. On the other hand, SILVA, with almost 9 million reference sequences, possesses a biannual updated comprehensive coverage of both small and large subunit ribosomal RNA sequences (**S3 Table**) [27].

Furthermore, Peng chose to use DADA2 for filtering and denoising raw reads, whereas Pi and Cheng opted for the QIIME tool. DADA2 is recognized for its ability to model and correct Illumina-sequenced amplicon errors and detect single-nucleotide polymorphisms (SNPs) across similar species. On the other hand, QIIME is an integrated pipeline that permits analysis of raw sequencing data, diversity, and taxonomic composition, and it is widely used due to its ability to seamlessly integrate various microbiome-specialized algorithms and databases, **S4 Table** [25, 27].

While striving for accuracy in gut microbiota identification, every methodology contends with unique challenges, including selecting sequencing regions, analytical pipelines, and choice of reference libraries or sequence databases that can introduce variations in study outcomes (**S1 Fig** and **S3 Table**). Variations in microbial identification were observed based on the sequencing methodology; Cheng et al. applied 16S rRNA sequencing to identify 9,708 OTUs across 10 genera [23]. On a different approach, Pi et al. used shotgun metagenomic sequencing, uncovering 72 CAGs, 19 genera, and 11 species [20]. Peng et al., employing 16S rRNA sequencing and shotgun metagenomics, discerned 3,852 OTUs across 58 genera and subsequently predicted differential metabolic pathways and 18 differential species, respectively (**S1 Fig**) [24]. It is imperative for future research to consider the significant influence of sequencing techniques on microbiota characterization, enhancing the accuracy and comprehensiveness of microbiota profiling analyses.

## Assessment of risk of bias

Koptez et al. and Pi et al. were judged 'Fair,' signifying scores of 5 to 10 on a scale of 14 in response to NIH Quality Assessment Tools, **Fig 4** and **Table 2**. Peng et al. and Cheng et al. garnered a 'Good' rating, with scores between 11 and 14 out of 14 questions, **Fig 4** and **Table 2**. Upon evaluating the studies included in this systematic review, several potential sources of bias were identified, affecting the study's internal validity. These biases arise from methodological and reporting limitations within the individual studies, including the lack of a thorough description and clearly defined study population sample size, incomplete or ambiguous

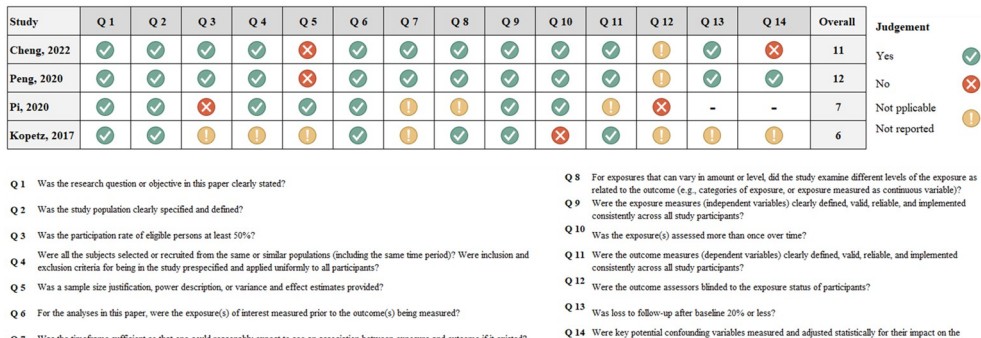

| Study | Q 1 | Q 2 | Q 3 | Q 4 | Q 5 | Q 6 | Q 7 | Q 8 | Q 9 | Q 10 | Q 11 | Q 12 | Q 13 | Q 14 | Overall |
|---|---|---|---|---|---|---|---|---|---|---|---|---|---|---|---|
| Cheng, 2022 | ✓ | ✓ | ✓ | ✓ | ✗ | ✓ | ✓ | ✓ | ✓ | ✓ | ✓ | ! | ✓ | ✗ | 11 |
| Peng, 2020 | ✓ | ✓ | ✓ | ✓ | ✗ | ✓ | ✓ | ✓ | ✓ | ✓ | ✓ | ! | ✓ | ✓ | 12 |
| Pi, 2020 | ✓ | ✓ | ✗ | ✓ | ✓ | ✓ | ! | ! | ✓ | ✓ | ! | ✗ | - | - | 7 |
| Kopetz, 2017 | ✓ | ✓ | ! | ! | ! | ✓ | ! | ✓ | ✓ | ✗ | ✓ | ! | ! | ! | 6 |

Judgement
- ✓ Yes
- ✗ No
- ! Not applicable
- ! Not reported

**Q 1** Was the research question or objective in this paper clearly stated?

**Q 2** Was the study population clearly specified and defined?

**Q 3** Was the participation rate of eligible persons at least 50%?

**Q 4** Were all the subjects selected or recruited from the same or similar populations (including the same time period)? Were inclusion and exclusion criteria for being in the study prespecified and applied uniformly to all participants?

**Q 5** Was a sample size justification, power description, or variance and effect estimates provided?

**Q 6** For the analyses in this paper, were the exposure(s) of interest measured prior to the outcome(s) being measured?

**Q 7** Was the timeframe sufficient so that one could reasonably expect to see an association between exposure and outcome if it existed?

**Q 8** For exposures that can vary in amount or level, did the study examine different levels of the exposure as related to the outcome (e.g., categories of exposure, or exposure measured as continuous variable)?

**Q 9** Were the exposure measures (independent variables) clearly defined, valid, reliable, and implemented consistently across all study participants?

**Q 10** Was the exposure(s) assessed more than once over time?

**Q 11** Were the outcome measures (dependent variables) clearly defined, valid, reliable, and implemented consistently across all study participants?

**Q 12** Were the outcome assessors blinded to the exposure status of participants?

**Q 13** Was loss to follow-up after baseline 20% or less?

**Q 14** Were key potential confounding variables measured and adjusted statistically for their impact on the relationship between exposure(s) and outcome(s)?

**Fig 4. Traffic light plot illustrating the risk of bias, assessed using the National Institutes of Health (NIH).** The colors green, yellow, and red represent low, unclear, and high risk of bias respectively in each domain/question of the included studies.

reporting of recruitment details, and inconsistency in applying the inclusion and exclusion criteria. Furthermore, the non-availability of raw data repositories was a notable limitation, hindering a comprehensive and transparent meta-analysis.

Only the investigation spearheaded by Peng et al. and colleagues considered all significant potential confounding variables. Cheng et al. conducted an exploratory study without statistical adjustments commonly used to balance the likelihood of chance findings in multiple comparisons. The aim was to find new insights and trends rather than validating hypotheses, emphasizing generating possible research directions over drawing statistically significant definitive conclusions.

## Discussion

A PubMed search using key terms "gut microbiota," "colorectal cancer," and "therapy" in a broad sense returns over 1,000 results, showcasing the expansive research in this area. Nevertheless, the volume of human clinical studies focused on CRC, gut microbiota, and ICIs is surprisingly deficient [16, 19].

Several studies have shown that the abundance of *Lachnospiraceae* was associated with favorable clinical outcome to ICIs in melanoma and hepatocellular carcinoma (HCC) patients [28–31]. Consistently, Cheng et al. reported a significant abundance of *Lachnospiraceae* in pan-cancer, including CRC, ICIs responders [23]. Peng et al. found a significant abundance of *Lachnospiraceae* in CRC ICIs responders, [24] underlining the potential role of this family of bacteria in enhancing the clinical efficacy of ICIs.

Both Peng et al. and Pi et al. reported an enrichment of *Akkermansia* and *Akkermansia municiphila*, respectively, in CRC patients demonstrating favorable outcomes from ICIs treatment [20, 24]. Similar ICIs' positive outcomes associated with *Akkermansia* were reported in non-small cell lung cancer (NSCLC) (35) and hepatocellular carcinoma (HCC) patients [32, 33]. Notably, *Akkermansia municiphila* was associated with favorable ICIs outcomes across multiple cancers, including NSCLC, [34] HCC, [35] and metastatic renal cell carcinoma (mRCC) [36]. There's only one study reporting the presence of *Akkermansia municiphila* in metastatic castrate-resistant prostate cancer (mCRPC) non-responders when ICIs therapy was combined with hormone therapy, namely enzalutamide [37].

Pi et al. found an increased presence of *Parasutterella* in CRC patients treated with anti-PD-1, irrespective of their ultimate clinical outcome [20]. Despite this bacteria's associations with various health conditions, [38] there is still no thorough clinical research validating its role in enhancing the effectiveness of PD-1-targeted treatments [39].

Peng et al. observed that CRC patients who respond to ICIs display increased bacterial activity, producing short-chain fatty acids (SCFA), such as acetate, propionate, and butyrate [24]. These SCFAs, fermented by gut bacteria like *Akkermansia*, *Bifidobacterium*, *Ruminococcus*, *Blautia*, *Bacteroides*, *Prevotella*, *Eubacterium rectale*, *Faecalibacterium*, [20, 23, 24] contribute to ICIs effectiveness by both providing energy to immune cells, such as B cells, memory T cells, and effector T cells via metabolic pathways like glycolysis, the Krebs cycle, and β-oxidation, and enhancing the antitumor immune response [40]. For instance, butyrate elevates CD8$^+$ T cell antitumor potency by promoting ID2 expression through IL-12 signaling. Moreover, valeric and butyric acid stimulate the expression of effector molecules like IFNγ and TNFα, increasing the antitumor impact of CTLs [40].

Current research indicates that in non-responders to ICIs, microbial activity is characterized by excessive lipopolysaccharide biosynthesis and sugar degradation [24]. These processes reportedly contribute to disturbed intestinal microecology, which in turn may inhibit antitumor immunity [24, 40].

The complexity of gut microbiota-ICI interactions, demonstrated in the reviewed studies, [19, 20, 23, 24] emphasize the importance, detail variation, and limitations of microbiota identification tools such as SILVA, GreenGenes, and MetaPhlAn2, **S3 Table**. Depending on the tool utilized, the accuracy and depth of taxonomic classifications, from genus to species or even strain level, can affect profiling accuracy due to variations in database updating, completeness, and computational methods. Furthermore, using different reference databases can influence study results; hence, specific microbiota's role in ICI efficacy might go unnoticed. Therefore, methodological standardization and alignment should be considered when comparing microbiota-immunotherapy research outcomes.

## Strengths and limitations

The reviewed studies [19, 20, 23, 24] exhibit robustness through rigorous methodologies, providing substantial evidence of a connection between gut microbiota and immune checkpoint inhibitors (ICIs) across various cancers. These studies contribute significantly to cancer treatment research by offering valuable insights gained through microbial identifications. However, it is essential to acknowledge the limitations of these studies, including small sample sizes, single-center origins, uncontrolled variability, and brief observational periods. Additionally, the unequal distribution of responders and non-responders across the studies poses a challenge to drawing comprehensive conclusions.

While Cheng and Kopetz's study employed population selection based on histopathological examinations, Peng and Pi did not clearly outline the criteria for selecting cancer patients. Furthermore, except for Cheng et al., all studies in this systematic review predominantly relied on baseline gut microbiota profiling without considering the dynamic nature of the microbiome. The lack of continuous assessment of microbiome changes throughout ICI treatment limits the generalizability of findings. Consequently, further comprehensive research is essential to establish more definitive conclusions.

The four studies included in this systematic review focused on colorectal cancer (CRC) patients, with 54% from China and the remaining 46% from various countries (Australia, Belgium, Canada, France, Ireland, Italy, Spain, and the United States), as indicated in **Table 1**. However, the predominance of data from one geographical area raises concerns about potential geographic bias and the influence of ethnocultural factors, given the likelihood of a single ethnic, dietary, and cultural background. To enhance the validity and reliability of findings, future research in this oncology and microbiota interplay domain must consider a more geographically diverse patient population, addressing these potential biases. This

recommendation considers the inherent challenges and complexities associated with designing and executing clinical studies involving cancer patients and collecting stool samples while accounting for a wide range of variables.

The systematic review's limitations revolve around excluding non-English language publications, increasing the risk of language bias. Additionally, this review may exhibit publication bias, as it only includes published studies, potentially omitting relevant data. The limited number of studies in this niche area and the variability among patient populations and cancer types in the included studies may have further confounded the results.

## Conclusion

The reviewed studies by Cheng et al., Peng et al., Pi et al., and Kopetz et al. underscored the pivotal role of gut microbiota in determining the effectiveness of immune checkpoint inhibitors in colorectal cancer treatment. Observations emphasized distinct disparities in richness and microbial composition between responders and non-responders. Notably, certain bacterial species and families such as *Faecalibacterium*, *Bifidobacterium*, *Lachnospiraceae*, *Akkermansia* sp., *Ruminococcaceae*, and *Prevotellaceae* appeared regularly as potential predictors of immunotherapeutic responses. In conclusion, additional research is required to validate these results and further investigate microbiota's broader implications on colorectal cancer ICI treatment.

## Supporting information

**S1 Fig. Bioinformatics workflows and analytical approaches in reviewed studies.**
(TIF)

**S1 Table. Updated search for "The role of microbiota in immunotherapy outcomes in colorectal cancer patients".**
(PDF)

**S2 Table. PRISMA 2020 checklist: Recommended items to address in a systematic review.**
(PDF)

**S3 Table. Overview of bioinformatics databases utilized for taxonomic identification in the reviewed studies.**
(PDF)

**S4 Table. Overview of bioinformatics tools utilized for functional pathway prediction in the reviewed studies.**
(PDF)

**S5 Table. Comparison of identified microbiota taxa in responders vs. non-responders across different tumor sites, treatment regimens, and time points in the reviewed studies.**
(PDF)

**S6 Table. Microbiota taxa utilized in the phylogenetic tree construction in Fig 2.**
(PDF)

## Acknowledgments

We express our gratitude to the National Medical Library team for their invaluable contribution and assistance, and we thank the Website team at mindthegraph.com for their technical assistance during the creation of Fig 3.

## Author Contributions

**Conceptualization:** Suad Mohamed Ajab, Sumaya Hasan Zoughbor, Linda Mari Östlundh, Zakeya Al Rasbi.

**Data curation:** Suad Mohamed Ajab, Sumaya Hasan Zoughbor, Lena Abdulbaset Labania, Marie Antonette Olanda, Obaid Alkaabi, Shamma Hamad Alkuwaiti, Shaikha Mohammed Alnuaimi, Zakeya Al Rasbi.

**Formal analysis:** Suad Mohamed Ajab, Lena Abdulbaset Labania.

**Funding acquisition:** Zakeya Al Rasbi.

**Investigation:** Zakeya Al Rasbi.

**Methodology:** Suad Mohamed Ajab, Sumaya Hasan Zoughbor, Linda Mari Östlundh, Zakeya Al Rasbi.

**Project administration:** Marie Antonette Olanda, Zakeya Al Rasbi.

**Resources:** Linda Mari Östlundh, Zakeya Al Rasbi.

**Supervision:** Linda Mari Östlundh, Zakeya Al Rasbi.

**Validation:** Sumaya Hasan Zoughbor, Lena Abdulbaset Labania, Linda Mari Östlundh, Hiba Salaheldin Orsud, Marie Antonette Olanda, Obaid Alkaabi, Shamma Hamad Alkuwaiti, Shaikha Mohammed Alnuaimi.

**Visualization:** Sumaya Hasan Zoughbor, Hiba Salaheldin Orsud, Marie Antonette Olanda, Obaid Alkaabi, Shamma Hamad Alkuwaiti, Shaikha Mohammed Alnuaimi.

**Writing – original draft:** Suad Mohamed Ajab, Hiba Salaheldin Orsud.

**Writing – review & editing:** Sumaya Hasan Zoughbor, Lena Abdulbaset Labania, Hiba Salaheldin Orsud, Obaid Alkaabi, Shamma Hamad Alkuwaiti, Shaikha Mohammed Alnuaimi, Zakeya Al Rasbi.

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
