## [Decision Letter · Decision Letter 0]

21 May 2024

PONE-D-24-10055Microbiota composition effect on immunotherapy outcomes in colorectal cancer patients: a systematic reviewPLOS ONE

Dear Dr. Al Rasbi,

Thank you for submitting your manuscript to PLOS ONE. After careful consideration, we feel that it has merit but does not fully meet PLOS ONE’s publication criteria as it currently stands. Therefore, we invite you to submit a revised version of the manuscript that addresses the points raised during the review process.

We look forward to receiving your revised manuscript.

Kind regards,

Vinod Kumar Yata, PhD

Academic Editor

PLOS ONE

Journal Requirements:

2. We note that Figure 3 in your submission contain copyrighted images. All PLOS content is published under the Creative Commons Attribution License (CC BY 4.0), which means that the manuscript, images, and Supporting Information files will be freely available online, and any third party is permitted to access, download, copy, distribute, and use these materials in any way, even commercially, with proper attribution. For more information, see our copyright guidelines: http://journals.plos.org/plosone/s/licenses-and-copyright.

a. You may seek permission from the original copyright holder of Figure 3 to publish the content specifically under the CC BY 4.0 license. 

Additional Editor Comments:

This systematic review highlights the potential of specific gut microbiota compositions, such as Akkermansia muciniphila and Eubacterium rectale enrichment, to predict better outcomes of immune checkpoint inhibitor therapy in colorectal cancer (CRC) patients. It underscores the need for larger, multicenter studies to validate these findings and advance personalized CRC treatment. We appreciate the effort you have put into this manuscript and believe that addressing the reviewers' comments will significantly improve its quality and impact. Please revise the manuscript accordingly and resubmit it for further consideration

Reviewers' comments:

Reviewer's Responses to Questions

**Comments to the Author**

1. Is the manuscript technically sound, and do the data support the conclusions?

Reviewer #1: Yes

2. Has the statistical analysis been performed appropriately and rigorously? 

Reviewer #1: Yes

3. Have the authors made all data underlying the findings in their manuscript fully available?

Reviewer #1: Yes

4. Is the manuscript presented in an intelligible fashion and written in standard English?

Reviewer #1: Yes

5. Review Comments to the Author

Reviewer #1: Thanks for the opportunity to review the manuscript titled " Microbiota composition effect on immunotherapy outcomes in colorectal cancer patients: a systematic review ".

The manuscript investigates whether composition of gut microbiota could potentially serve as a biomarker for predicting the clinical effectiveness of immune checkpoint inhibitors in CRC patients. Phyla including Firmicutes, Bacteroidata, Actinobacteria, and Verrucomicrobiota were associated with CRC patients’ clinical response toward ICIs treatment. Ruminococcaceae and few more families of gut bacteria were found to be differentially related to clinical response of CRC patients. This research concludes that there is need for additional comprehensive, multicentre studies with larger sample sizes to validate reported microbiota and expand the understanding of the role of gut microbiota in CRC for ICIs therapy. The study's design incorporates suitable method to achieve the outlined objective. The content is logically structured, providing cohesive information.

The manuscript can be considered for publication. However, some points need to be addressed. Authors should explain how their manuscript is different from other such publications linking the gut microbes with colorectal cancer. Some more comments have been appended in the attached file. I request the authors to update their important manuscript on the basis of the raised comments and submit an updated version.

Thanks for the opportunity to review this submission.

6. PLOS authors have the option to publish the peer review history of their article (what does this mean?). If published, this will include your full peer review and any attached files.

Reviewer #1: No

---

## [Author Response · Author response to Decision Letter 0]

24 Jun 2024

June 05, 2024

Vinod Kumar Yata

Academic Editor

PLOS ONE

Ref PONE-D-24-10055: Microbiota composition effect on immunotherapy outcomes in colorectal cancer patients: a systematic review

Dear Dr. Vinod Kumar Yata 

We appreciate the feedback from the reviewers and the opportunity to submit a revised version of our manuscript. Following the comments provided, we have thoroughly revised the manuscript to incorporate the suggested amendments. For ease of review, the revised manuscript was displayed in track mode, and some paragraphs were highlighted, addressing some of the reviewer comments. Below, detailed responses to each of the reviewers’ comments are provided.

Journal Requirements

Comment 1: Please ensure that your manuscript meets PLOS ONE’s style requirements, including those for file naming.

Answer: 

We confirm that our manuscript adheres to the style guidelines of PLOS ONE.

Comment 2: We note that Figure 3 in your submission contains copyrighted images. All PLOS content is published under the Creative Commons Attribution License (CC BY 4.0), which means that the manuscript, images, and Supporting Information files will be freely available online, and any third party is permitted to access, download, copy, distribute, and use these materials in any way, even commercially, with proper attribution. We require you to either (1) present written permission from the copyright holder to publish these figures specifically under the CC BY 4.0 license, or (2) remove the figures from your submission:

Answer: 

The images in the manuscript were reproduced through the utilization of mindthegraph.com for Figure 3 and PowerPoint of Figure S1 in the format of PNG. Subsequently, a CC BY 4.0 license was acquired in response to a specific request.

Reviewers’ comments

From Methods Section:

Comment 1: Authors could elaborate on the rationale behind the decision to include only published observational studies and exclude experimental animal studies, reviews, comments, case reports, editorials, and conference abstracts.

Answer: 

Thank you for your comment. As stated on Page 5, Paragraph 3 of our manuscript, the intent is to shed light on the association between gut microbiota and the clinical efficacy of immune checkpoint inhibitors in colorectal cancer patients, thus directly contributing to the clinical oncology community’s body of knowledge. Our research aims to provide a comprehensive overview of the existing understanding regarding the specified objective, with the intention of enhancing the methodologies employed by healthcare practitioners and researchers, consequently leading to favorable advancements in therapeutic approaches within cancer therapy.

To ensure the focus remained on applicable, human-centric data, we exclusively considered peer-reviewed published observational studies animal experiments were omitted from consideration due to the potential limitations in directly extrapolating findings to human subjects. Similarly, reviews, comments, case reports, editorials, and conference abstracts were not included; while these can offer valuable insights, they do not constitute the original research needed for our analysis. Reviews and comments typically summarize other findings, case reports provide anecdotal evidence with limited generalizability, editorials represent opinions, and conference abstracts may present unvetted preliminary data, apart from one abstract was included which was part of the international CheckMate 142 clinical trial. Our stringent selection criteria were applied to construct an analysis founded upon the most reliable and broadly applicable evidence.

Please refer to the revised Methods section on Page 7 for details on the amendments made.

Comment 2: Authors could elaborate on the methodologies used by some most cited studies to analyze the differential observations regarding gut microbiota composition of colorectal cancer patients undergoing immunotherapy. 

Answer: 

Thank you for your valuable comment. We recognize the importance of the methodologies used in the primary studies for analyzing gut microbiota composition among colorectal cancer patients undergoing immunotherapy. As part of our systematic review criteria, we stipulated that gut microbiota composition must be reported via molecular analysis, which we have detailed in the Methods section (page 6 Line 127).

Furthermore, we dedicated a section in our Results titled “Comparative Analysis of Methodologies Utilized in Microbiota Profiling,” (Page 21) wherein we critically assess the methodologies employed by the studies included in our review. This section is supported by relevant table S3,S4 and Figure S1 which systematically showcase the fundamental comparisons and contrasts between the reported methods in each paper. This includes the molecular techniques utilized for the profiling of microbiota, and the bioinformatic tools applied for analysis.

We believe this approach provides a comprehensive overview of the methodologies and allows readers to understand the potential impact of these methods on the study outcomes. However, if additional elaboration on specific methodologies used by the most cited papers is required, we are willing to provide further details and discussion as per the journal’s and the reviewers’ recommendations.

Comment 3: Were there any notable differences in patient recruitment strategies among the included studies, and how might these differences impact the generalizability of their findings?

Answer: 

Thank you for bringing attention to the recruitment strategies of the included studies. As detailed in Table.1 and the “Study Characteristics” section (page 14), we meticulously cataloged the inclusion and exclusion criteria for each study. Moreover, in the “Assessment of Risk of Bias” section (page 23), we discussed methodological and reporting limitations within the individual studies, including the lack of a thorough description and clearly defined study population sample size, as well as incomplete or ambiguous reporting of recruitment details, and inconsistency in applying the inclusion and exclusion criteria.

This careful documentation allowed us to identify the “strengths and limitations” of the studies (page 28). Notably, the four studies included in this systematic review focused on colorectal cancer patients, with 54% from China and the remaining 46% from various countries, as indicated in Table 1. However, the predominance of data from one geographical area raises concerns about potential geographic bias and the influence of ethnocultural factors, given the likelihood of a singular ethnic, dietary, and cultural background.

To address the reviewer’s question, these differences in recruitment approaches were carefully considered in relation to their impact on the generalizability of the study findings, as also detailed in the “Strengths and Limitations” section (Page 28). We recommended, “To enhance the validity and reliability of findings, future research in this domain must consider a more geographically diverse patient population, addressing these potential biases.” (Page 28-29) This recommendation takes into account the inherent challenges and complexities associated with designing and executing clinical studies involving cancer patients and collecting stool samples while accounting for a wide range of variables.

Comment 4: The article could provide a comprehensive overview of the role of phyla-based gut microbiota effects in colorectal cancer and its response to immunotherapy.

Answer:

Thank you for your positive feedback on the scope of our manuscript. We appreciate your recognition of our efforts to provide a comprehensive overview of phyla-based gut microbiota effects in colorectal cancer and its response to immunotherapy.

From Results Section: 

Comment 1: Can you elaborate on the characteristics of the studies included in the analysis, such as their sample sizes, publication years, and geographic locations?

Answer: 

Thank you for your inquiry. We have detailed the sample sizes, publication years, and geographic locations of the studies within the manuscript. Specifically, this information is presented in Table 1 and described in the text on page 14. To elaborate:

• The studies were published between 2017 and 2022.

• The sample sizes ranged from 60 to 89 participants.

• Cheng et al. 2022, Peng et al. 2020, and Pi et al. 2020 were conducted in China, while the study by Koptez et al. 2017 is part of the international CheckMate 142 clinical trial, which took place across eight countries: Australia, Belgium, Canada, France, Ireland, Italy, Spain, and the United States.

We appreciate your request for clarification, and we believe that the information provided in our manuscript will satisfy the need for a clear and comprehensive understanding of the study characteristics (Page 14).

Comment 2: How did the studies by Cheng et al., Peng et al., and Kopetz et al. differ in their methodologies and findings regarding the relationship between gut microbiome composition and response to immunotherapy in colorectal cancer patients (except the clinical trials)?

Answer: 

We appreciate the request for further clarification. These details are outlined in the results section “Predictive taxonomic profiles of CRC patients in response to ICIs” on pages 15-21. To summarize:

Kopetz et al. observed shifts in the gut microbiome of CRC patients with dMMR and high MSI-H treated with ICIs, identifying lower abundances of Rothia and Rothia mucilaginosa in partial responders compared to those with progressive disease. This study served as part of the broader CheckMate 142 clinical trial series.

Pi et al. investigated the presence of different bacterial co-abundant gene groups between responders and non-responders to CRC ICIs. Responders with six months of progression-free survival demonstrated a high abundance of Akkermansia muciniphila. Conversely, non-responders were found to harbor different bacterial CAGs, including several Firmicutes and Bacteroides families.

Peng et al. implemented 16S rRNA gene sequencing in advanced-stage gastrointestinal cancer patients to identify differential abundant operational taxonomic units between responders and non-responders to ICIs. They found an abundance of Prevotellaceae, potentially predictive of disease stabilization, and active machine learning techniques identified Firmicutes taxa as top predictors for therapy response. Their research also confirmed a consistent correlation between Prevotella and Bacteroides.

Cheng et al. conducted a comprehensive study across various types of advanced cancers. They noted distinct differences in gut microbial biodiversity between responders and non-responders before and after ICI therapy, recording an abundance of certain microbes like Archaea and Lentisphaerae in responders. In their network analysis, they determined the interrelationships among bacterial phyla within the gut microbiota of advanced cancer patients.

These studies, while utilizing similar sequencing techniques, differed in patient populations, sequencing analysis, prediction methods, and the types of ICIs employed; thus, providing a multifaceted picture of how gut microbiota composition might influence immunotherapy’s effectiveness in CRC.

We hope this synthesis of methodologies and findings provides a more comprehensive understanding of the intricate relationship between the gut microbiome and immunotherapy outcomes.

Comment 3: What were the primary outcomes regarding the predictive taxonomic profiles of colorectal cancer patients in response to immune checkpoint inhibitors (ICIs) based on the included studies?

Answer: 

We appreciate your attention to the details of our manuscript. This question is closely related to the previous question that focused on how the studies by Cheng et al., Peng et al., and Kopetz et al. differ in their methodologies and findings.

To reiterate and address the current question, the primary outcomes highlighted in these studies all demonstrated that there are specific taxonomic profiles of microbiota that may predict the effectiveness of ICIs in colorectal cancer patients. The key findings across the included studies are as following:

• Kopetz et al. found that certain taxa, such as Rothia and Rothia mucilaginosa, were less abundant in CRC patients who showed partial responses to ICIs. These differences in abundance may serve as indicators of patient responses to ICIs.

• Pi et al. and Peng et al. both noted a higher abundance of Akkermansia muciniphila in responders, implying a potential predictive role for this bacterium. Peng et al. also found Prevotellaceae to be more abundant in responders than in non-responders.

• Cheng et al. observed that specific microbes such as Archaea, Lentisphaerae, and other taxa were more abundant in pan-cancer responders prior to therapy, indicating a possible pan-tumor predictive signature for ICI response.

Each of these studies contributes to a growing body of evidence that suggests a significant association between the gut microbiome composition and the clinical efficacy of ICIs in colorectal cancer therapy. Their outcomes suggest that certain microbial signatures could potentially serve as biomarkers for predicting immunotherapy responses.

We have covered these outcomes extensively in the manuscript, particularly in the Results and Discussion sections, which we believe will provide comprehensive insights for the readers.

Regarding the Bias Section: 

Comment 1: Can the authors explain the potential sources of bias identified in the included studies, such as methodological and reporting limitations?

Answer: 

Thank you for your question. We have evaluated the risk of bias in each study and have provided detailed assessments within our manuscript “Assessment of risk of bias” on pages 23-24. Here are some key points from our evaluation:

Kopetz et al. and Pi et al. were rated as ‘Fair’ according to the NIH Quality Assessment Tools, which suggests that these studies have certain methodological limitations that could introduce bias. These limitations include a lack of comprehensive descriptions of the study population, ambiguous reporting of recruitment details, and inconsistencies in the application of inclusion and exclusion criteria. Furthermore, the absence of accessible raw data repositories presents an additional challenge, restricting the ability to perform a full and transparent meta-analysis.

On the other hand, Peng et al. received a ‘Good’ rating, indicating that they addressed most of the potential confounding factors at play, which lessens the risk of bias. Cheng et al. conducted an exploratory study and did not apply the usual statistical adjustments to limit the possibility of finding chance associations in the course of making multiple comparisons. The goal of their study was to uncover new trends and generate hypotheses for further research, rather than confirming hypotheses or drawing statistically significant definitive conclusions.

These potential sources of bias have been considered and discussed in our systematic review. We have transparently reported these limitations and considered them when interpreting the results of our systematic review. This assessment ensures that readers can critically evaluate the validity and reliability of the study findings presented.

For further details and the comprehensive risk of bias assessment for each included study, please refer to Figure 4 and Table 2 in the manuscript.

Comment 2: How did the authors address or mitigate the impact of these biases on the internal validity of the systematic review?

Answer: 

Thank you for the follow-up question. As outlined in our previous response, we conducted a rigorous evaluation of the risk of bias in each of the included studies using standardized assessment tools. To mitigate the impact of these biases:

• We provided a clear and comprehensive synthesis of the findings, considering the limitations and methodological quality of each study.

• We discussed the implications of these biases in our results and conclusions, ensuring our interpretation of the data is our analysis of the data is impartial and firmly rooted in the framework of these limitations.

• Where possible, we sought to validate key findings t

---

## [Editor Report · Decision Letter 1]

9 Jul 2024

Microbiota composition effect on immunotherapy outcomes in colorectal cancer patients: a systematic review

PONE-D-24-10055R1

Dear Dr. Al Rasbi,

We’re pleased to inform you that your manuscript has been judged scientifically suitable for publication and will be formally accepted for publication once it meets all outstanding technical requirements.

Kind regards,

Vinod Kumar Yata, PhD

Academic Editor

PLOS ONE

Additional Editor Comments (optional):

I agree with the authors' response and recommend this research paper for publication
---

## [Editor Report · Acceptance letter]

15 Jul 2024

PONE-D-24-10055R1 

PLOS ONE

Dear Dr. Al Rasbi, 

I'm pleased to inform you that your manuscript has been deemed suitable for publication in PLOS ONE. Congratulations! Your manuscript is now being handed over to our production team.

Kind regards, 

on behalf of

Dr. Vinod Kumar Yata 

Academic Editor

PLOS ONE